# Investigation of the Suitability of a Combination of Ethyl-Να-dodecanyl-L-arginat_HCl (LAE) and Starter Culture Bacteria for the Reduction of Bacteria from Fresh Meat of Different Animal Species

**DOI:** 10.3390/foods12224138

**Published:** 2023-11-15

**Authors:** Maike Drevin, Madeleine Plötz, Carsten Krischek

**Affiliations:** Institute of Food Quality and Food Safety, University of Veterinary Medicine Hannover, Foundation, Bischofsholer Damm 15, 30173 Hannover, Germanymadeleine.ploetz@tiho-hannover.de (M.P.)

**Keywords:** pork, beef, chicken meat, LAE, starter cultures, microbiology, color, myoglobin redox forms

## Abstract

Meat can be contaminated with (pathogenic) microorganisms during slaughter, dissection and packaging. Therefore, preservation technologies are frequently used to reduce the risk of (fatal) human infections due to the consumption of meat. In this study, we first investigated, if the application of ethyl-Nα-dodecanyl-L-arginate hydrochloride (LAE) and the starter culture bacteria *Staphylococcus carnosus* and *Lactobacillus sakei*, either single or in combination, influences the bacteria number on pork, chicken meat and beef, inoculated with *Brochothrix (Br.) thermosphacta* (all meat species) or *Salmonella* (*S*.) Typhimurium (pork), *Campylobacter* (*C*.) *jejuni* (chicken) and *Listeria* (*L*.) *monocytogenes* (beef), before packaging under modified atmosphere and on days 7 and 14 of storage. To evaluate effects of the treatment on the appearance during storage, additionally, the physicochemical parameters color and myoglobin redox form percentages were analyzed. LAE regularly resulted in a significant reduction of the number of all bacteria species on day 1 of storage, whereas up to day 14 of storage, the preservation effect did not persist in nearly all samples, except in the beef with *Br. thermosphacta*. However, with the starter culture bacteria on day 1, only *L. monocytogenes* on beef was significantly reduced. Interestingly, on day 7 of storage, this reducing effect was also found with *S*. Typhimurium on pork. *Br. thermosphacta*, which was principally not influenced by the starter culture bacteria. The combinatory treatment mainly resulted in no additional effects, except for the S. Typhimurium and *Br. thermosphacta* results on pork on day 7 and the *Br. thermosphacta* results on beef on day 14. The physicochemical parameters were not influenced by the single and combinatory treatment. The results indicate that LAE was mainly responsible for the antimicrobial effects and that a combination with starter culture bacteria should be individually evaluated for the meat species.

## 1. Introduction

Food of animal origin, especially fresh meat, has a limited shelf life. Fresh meat provides better growth conditions for many microorganisms. In addition to spoilage, diseases caused by pathogenic microorganisms such as *Campylobacter jejuni*, *Salmonella* spp. or *Listeria monocytogenes* pose a major risk in connection with the consumption of certain foods such as meat and dairy products [1,2]. Physical, chemical and biological preservation methods like drying, heating, freezing, chilling, freeze-drying, salting and smoking are used to preserve foods [3]. However, for fresh meat, only a few preservation methods are permitted by law. Besides cooling and freezing (not for fresh poultry meat), packaging under modified atmosphere or vacuum is allowed for fresh meat [4]. However, other (not legally allowed) preservation methods for fresh meat have been regularly investigated. These are, for example, irradiation with UV-C (physical process) [5,6], treatment with ethyl-Nα-dodecanyl-L-arginate hydrochloride (LAE) or peroxyacetic acid (chemical process) [7,8,9,10] or application of specific mixtures of microorganisms (starter culture bacteria, or biological process) [11,12,13].

LAE is a chemical substance which is approved in the European Union (EU) as a food additive for heat-treated meat products and it has a spectrum of antimicrobial activity against Gram-positive and Gram-negative bacteria, yeasts and molds [14,15] and is generally recognized as safe (GRAS), as it is rapidly converted to L-arginine ethyl ester via the cleavage of lauroyl side chain or Nα-lauroyl-L-arginine (LAS) via the loss of the ethyl ester. The resulting intermediates are hydrolyzed to form L-arginine, which is further metabolized to urea and ornithine. Lauric acid is a saturated fat widely found in many vegetable fats and can enter normal fatty acid metabolism [16]. Although LAE is not approved for the preservation of fresh meat in the EU, scientific studies have been published showing antimicrobial effects of LAE on fresh meat [8,9,17,18,19,20,21,22,23,24,25,26].

Starter cultures are regularly used for the production of foods such as yoghurt, cheese or raw fermented sausage. The addition of these starter cultures leads, for example, to the acidification of the food (through the formation of lactic acid) and the formation of antimicrobial substances such as bacteriocines. Starter culture mixtures are used which, depending on the product, are composed of certain microorganisms. These microorganisms must be identifiable, biologically pure as well as hygienically safe and have sufficient stability of the mentioned properties. Unlike in sausage, yoghurt or cheese production, where the application of starter cultures is allowed and where these bacteria are actively mixed with the meat or milk during production, the application is not allowed in fresh meat, as already described above. However, because of the advantages of these bacterial mixtures, many studies have been published showing antimicrobial effects of starter cultures after application to fresh meat [11,12,13,27,28,29].

Considering climate change and the expected increase in temperature, which can influence the general microbial growth, it is not clear if and how this temperature alteration has an impact on the microbiological safety of food. If a temperature effect is to be expected, it is urgently necessary that the policy makers/competent authorities and the food industry are prepared and act quickly [30]. This includes, for example, the fast approval of new preservation technologies and the introduction and application of these methods in the food industry. The application should also consider the combination of different (new) preservation technologies [3].

As previous studies often only investigated the antimicrobial effects of one preservation method, the aim of this study was to investigate the effect of single and combined treatment of LAE and starter cultures on the safety of fresh meat of different slaughter species. Therefore, samples of beef, pork and chicken meat were first inoculated with different specific bacteria species, pathologically relevant for the meat species (e.g., *Salmonella* Typhimurium on pork), and with the spoilage bacteria species *Brochothrix thermosphacta*. After the inoculation, the meat was treated with LAE and a starter culture mixture individually and in combination and packed in modified atmosphere packages (MAP). The MAP were analyzed microbiologically and physicochemically on days 0, 7 and 14 of chilled storage. The physicochemical analyses were performed to additionally evaluate if the treatment alters the appearance of the meat, as this may limit the use of the preservation method individually and in combination.

## 2. Materials and Methods

### 2.1. Material

The experiments were performed in triplicate (N = 3). For the storage tests, cutlets of pork (topside), beef (topside) and chicken meat (breast) from different batches, stored in MAP, were purchased from a local supermarket on the day of the experiment and stored at 4 °C. For the inoculation experiments with *Brochothrix* (*Br*.) *thermosphacta* (beef, pork, chicken), *Listeria* (*L*.) *monocytogenes* (beef), *Salmonella* (*S*.) Typhimurium (pork) and *Campylobacter (C.) jejuni* (chicken), meat pieces with a size of 20.25 cm^2^ were cut from the center of the meat. For the physicochemical analyses, additional cutlets of the pork, beef and chicken meat were used. The latter samples were not inoculated with the bacteria before treatment.

### 2.2. Treatment and Sample Processing

For the inoculation experiments, *Brochothrix thermosphacta* (German Collection of Microorganisms and Cell Cultures (DSM) 20171, beef, pork, chicken), *Listeria monocytogenes* (DSM 20600, beef), *Salmonella* Typhimurium (DSM 19587, pork) and *Campylobacter jejuni* (DSM 4688, chicken) were used. For the treatment with the starter cultures, *Staphylococcus carnosus* (DSM 20501) and *Lactobacillus sakei* (DSM 6333) were used. Before the experiment, the bacteria species for the inoculation and for the (starter culture) treatment were plated on Columbia blood agar with sheep blood (Oxoid GmbH, Wesel, Germany) and incubated for 24 h at 37 °C. Colonies from the agar were transferred to sterile saline (0.9% NaCl) before adjusting the solution to McFarland turbidity standards of 3.0 using a densimat (BioMérieux SA France IDN 013615, Craponne). The total bacteria number in the solution after McFarland turbidity analysis was always analyzed on the inoculation day, as described in 2.5. The bacteria suspension was diluted before inoculation/mixture and treatment (starter cultures) to achieve an appropriate bacteria concentrations of 10^6^ cfu/mL (inoculated bacteria) or 10^7^ cfu/mL (each starter culture bacteria). The starter culture bacteria were mixed (1:1) to achieve a concentration before application of 0.5 × 10^7^ cfu/mL. For the inoculation, 100 µL of the bacteria suspension was applied to the meat surface and carefully spread with a spatula to achieve a final concentration (fin. conc.) of 10^5^ cfu on the meat. After an incubation period of 20 min, the meat samples were treated with LAE (VEDEQSA, Barcelona, Spain, stock solution 10% (100,000 ppm) LAE dissolved in 90% glycerol) after dilution to 40,000 ppm with 90% glycerol, the starter culture mixture or with a combination of LAE and the starter cultures. For the LAE treatment, 100 µL LAE (fin. conc.: 4000 ppm) and 100 µL NaCl (0.9%, solvent of the starter culture mixture), for the starter culture study, 100 µL of the bacteria mixture (fin. conc.: 0.5 × 10^6^ cfu/mL) and 100 µL glycerol (3600 ppm, solvent of the LAE) and for the combination 100 µL of the bacteria mixture (fin. conc.: 10^6^ cfu/mL) and 100 µL of the LAE (fin. conc.: 4000 ppm) were applied. In the control treatment group, 100 µL of NaCl (0.9%) and 100 µL glycerol (3600 ppm) were distributed on the meat surface. After application in all treatment groups, the solutions were again carefully spread with the spatula.

After treatment, the meat pieces were packed in modified atmosphere packages (MAP, 70% O_2_, 30% CO_2_) (packaging machine T100, Multivac Sepp Haggenmüller, GmbH & Co. KG, Wolfertschwenden, Germany) and stored at 4 °C up to day 14. The MAP were opened on days 0 (day of treatment and packaging), 7 and 14, and 25 g was removed for the microbiological investigations. Then, the color of the products was determined at the surface. A thin layer from the meat surface (0.5 to 1.0 cm) used for the color analysis was removed and cut into small squares (0.5 × 0.5 × 0.5 cm^3^). The squares were packed in plastic bags (SR-Beutel 135 × 180 90 µm, Dagema eG, Willich, Germany), frozen in liquid nitrogen and stored at a maximum of −70 °C until determination of the percentages of the myoglobin (Mb) redox forms. The color determination was carried out in the same way as the above treatment methods (LAE, NaCl, Glycerin), packed in modified atmosphere packages and also examined on days 0, 7 and 14.

### 2.3. Physical Parameters 

The color values lightness (L*), redness (a*) and yellowness (b*) were analyzed on the surface of the non-inoculated but treated meat pieces with a colorimeter (Minolta CR 400^®^, Konica-Minolta GmbH, Langenhagen, Germany, 2° Standard observer, D65 illuminant, 8 mm measuring field). For calibration of the colorimeter, a specific white plate (Y = 84.0, x = 0.3226, y = 0.3392, Konica-Minolta GmbH) was used. Color measurements were repeated at least five times.

### 2.4. Chemical Parameters

In order to analyze if the treatment not only influences the color, but also the percentages of the Mb redox forms, the meat squares were analyzed considering the publication of Kernberger-Fischer et al. [31]. Therefore, up to 5 g of the frozen samples were homogenized with a Polytron PT 2500 homogenizer (Kinematica GmbH, Luzern, Switzerland) at 15,000 U/min for 2 min in 10 mL phosphate buffered saline (PBS, pH 7.4) on ice, before the suspension was centrifuged for 30 min at 35,000 g (Sorvall RC 5 C Plus, Thermo Scientific Langenselbold, Germany) at 4 °C. After transfer of the supernatant to 2.5 mL cuvettes, the solution was measured photometrically (Evolution 201-UV–VIS-Spectrophotometer, Thermo Scientific) at wavelengths of 525, 503, 557 and 582 nm. For the calculation of the percentages of the oxymyoglobin (OxyMb) and metmyoglobin (MetMb), the equations presented by Tang et al. [32] were considered. All analyses were performed in triplicate.

### 2.5. Microbiological Parameters

For the microbiological analyses of the inoculated and packaged meat, the bacteria on the meat surfaces were carefully wiped at first with a wet (with sterile saline solution (1% peptone, 0.85% NaCl, pH 7.0)) and then with a dry swap. Both swaps were transferred to sterile saline solution and vortexed for 1 min to remove the bacteria from the swaps. Then, serial 10-fold dilutions up to 10^6^ were performed using again sterile saline solution.

For analysis of the bacteria 0.1 mL of the appropriate dilutions were pipetted onto the particular agar plate, depending on the bacteria species. The solution was then distributed on the agar with a sterile spatula. OCLA agar was used for analysis of *L. monocytogenes*, Streptomycin-Inosit-Neutralrot-Agar (SIN) agar for analysis of *Br. thermosphacta*, CCDA agar for *C. jejuni* and XLD agar for *S*. Typhimurium. The bacteria species were incubated as follows: *L. monocytogenes* and *S*. Typhimurium at 37 °C for 24 h; *Br. thermosphacta* at 25 °C for 24 h; and *C. jejuni* at 42 °C (5–6% O_2_, 10% CO_2_, 84–85% N_2_) for 24 h.

To analyze the total number of bacteria after McFarland turbidity analysis and before inoculation, serial 10-fold dilutions up to 10^7^ of the solution were performed using saline solution with peptone (0.85% NaCl and 0.1% peptone). An amount of 0.1 mL of the 10^5^, 10^6^ and 10^7^ dilutions were pipetted on the appropriate agar plates and incubated, as presented above. The bacteria concentration was always approximately 10^8^ cfu/mL.

If no colony was detected on the agar plates with the lowest dilution, half of the detection limit of 100 cfu/cm^2^ was considered for the statistical analysis (50 cfu/cm^2^; log_10_ 1.7 cfu/cm^2^).

### 2.6. Statistical Analysis

Statistical analysis was performed using SAS Enterprise Guide 7.1 (SAS Institute Inc., Cary, NC, USA) considering the following model:Y_ij_ = μ + T_j_ + R_j_ + ε_ij_
where Y_ij_ = observation value; μ = overall mean; T_j_ = fixed effect of the treatment; Rj = random effect of replication; ε_ij_ = random error.

All data were first tested for normal distribution (Shapiro–Wilk test) and variance homogeneity (Levene test). If the data were normally distributed and variance homogeneous, the data were statistically analyzed using ANOVA. If the results were not normally distributed and/or not variance homogeneous, the Kruskal–Wallis-test was used for the statistical analysis. Differences were considered significant if the *p*-value was below 0.05.

## 3. Results

### 3.1. Microbiological Results

#### 3.1.1. *Brochothrix thermosphacta*

In pork, the treatment with LAE and the combination of LAE and the starter cultures results on day 0 (packaging day) in a significant (*p* ≤ 0.05) reduction of the bacteria number in comparison to the control and starter culture treatment groups. During storage, up to day 7, the combination treatment significantly (*p* ≤ 0.05) reduces the number of *Br. thermosphacta* compared to the control samples. However, the results after treatment with LAE and the starter culture mixture were comparable with all other treatment groups. On day 14 of MAP storage, the bacteria numbers did not differ significantly (*p* > 0.05) between the treatment groups (Figure 1).

In chicken meat, the treatment only influences the number of *Br. thermosphacta* on the packaging day 0, with significant reductions (*p* ≤ 0.05) of the bacteria on the meat treated with LAE and the combination of LAE and starter cultures compared to the control and starter culture samples. After applying starter cultures, the bacteria results were similar to those of the control and combination samples (Figure 2).

With regard to the number of *Br. thermosphacta* inoculated on beef before treatment and packaging on day 0, LAE and combination samples showed significantly (*p* ≤ 0.05) lower values compared to the control samples, whereas the results of the meat, treated with starter cultures only, differed only significantly (*p* ≤ 0.05) from the combination samples. During further storage up to day 7, treatment with LAE and with LAE + starter cultures resulted in significantly (*p* ≤ 0.05) lower numbers of *Br. thermosphacta* compared to the results of both other treatment groups. On day 14 of storage beef samples treated with LAE, starter cultures and their combination after inoculation with *Br. thermosphacta* showed significantly (*p* ≤ 0.05) lower results than the control samples. Interestingly, the bacteria numbers of the LAE samples differed also significantly (*p* ≤ 0.05) from those of the combination samples, whereas samples treated with the starter culture only were comparable with both other treatment groups (Figure 3).

#### 3.1.2. *Salmonella* Typhimurium, *Campylobacter jejuni*, *Listeria monocytogenes*

The effects of the treatments on different regularly pathogenic microorganisms are presented in Figure 4, Figure 5 and Figure 6. Inoculation of pork with *S*. Typhimurium before treatment and packaging resulted on day 0 in significant reductions (*p* ≤ 0.05) of the bacteria numbers of the combination samples compared to the control probes, whereas the results of pork, treated with LAE and starter cultures only, were comparable results with the results of the other treatment groups. On day 7, the control samples had significantly (*p* ≤ 0.05) higher bacteria numbers than all other treatment groups, whereas the results of the combination samples were also lower (*p* ≤ 0.05) than those of the LAE and starter culture samples. The treatment groups showed a comparable number of *S*. Typhimurium on day 14 of MAP storage (Figure 4).

On chicken meat, treatment with LAE and LAE and starter cultures directly after inoculation and packaging (day 0) resulted in significant (*p* ≤ 0.05) reductions of the bacteria numbers in comparison to the control and starter culture samples. On day 7 of MAP storage, the latter described significance (*p* ≤ 0.05) could only be found in comparison to the control samples, whereas the bacteria number of the chicken meat, treated with starter culture only, were comparable with the results of all other treatment groups. On day 14 of storage, the bacteria results were all below the detection limit of 2.0 log_10_ cfu/cm^2^. Therefore, half the detection limit (1.7 log_10_ cfu/cm^2^) was shown in the figure (Figure 5).

Finally, inoculation of beef with *L. monocytogenes* resulted, directly after packaging in MAP (day 0) and on days 7 and 14, in significantly (*p* ≤ 0.05) higher bacteria numbers of the control samples compared to the results of all other treatment groups. On day 7, combination samples showed significantly (*p* ≤ 0.05) lower concentrations of Listeria than the starter culture samples. The LAE results were comparable with those of the latter samples. On day 7, the significant differences were nearly like those on day 0 with the exception that the LAE samples showed the lowest bacteria numbers differing significantly (*p* ≤ 0.05) from the results of the control and starter culture samples. On day 14, the beef samples of all treatment groups (except control samples) showed concentrations of *L. monocytogenes* below the detection limit and again the half detection limit of 1.7 log_10_ cfu/cm^2^ is presented in the figure. (Figure 6).

### 3.2. Physicochemical Results

Analysis of the color values of the different meat species depending on the treatment resulted in no significant differences on days 0 and 14 of MAP storage (Table 1).

Analysis of the percentages of the myoglobin redox forms OxyMb and MetMb of the different meat species depending on the treatment resulted in no significant differences on days 0 and 14 of MAP storage (Table 2).

## 4. Discussion

As far as we know, no studies that analyzed the antimicrobial and physicochemical effects of LAE and starter culture treatment in combination have been published so far. Therefore, we mainly discuss the published results of the single treatments with LAE and the starter culture bacteria.

The significant effects of LAE on the bacteria numbers on day 0 (directly after treatment and packaging) in the present study principally agree with publications that also analyzed the antibacterial impact of LAE using meat of different species [8,9,17,18,19,20,21,22,23,24,25,26]. Most of the studies inoculated the products with the bacteria before LAE treatment. However, the reduction of the bacteria number on meat is in the different studies (like in the present study) with 0.5 up to 2.0 log_10_ cfu quite low. This is in contrast to studies that investigated the effects of LAE in the reaction tube (in vitro), where higher antimicrobial effects at lower LAE concentrations could be obtained analyzing different bacteria species [9,14,17,19,20,33,34]. Reasons for this effect of the meat (matrix) might be the meat protein or the pH value, which change the growth properties of the bacteria and/or influence the degradation/inactivation of the LAE [35]. Additionally, the influence of the meat structure, which can protect the bacteria from the LAE, should also be considered. Anyway, the reasons for the antibacterial effects of the cationic surfactant LAE are due to the alteration of the morphology of the bacteria cells. This was shown in different scanning electron microscopic investigations with *E. coli* [14], *E. coli*, *L. monocytogenes* and *Br. thermosphacta* [33], *L. monocytogenes* and *E. coli* [34], *S*. Typhimurium and *S. aureus* [36], *E. coli*, *L. monocytogenes, Br. thermosphacta* and *Salmonella* spp. [37] or *E. coli* [38]. Pattanayaiying et al. [33] suggested that the LAE effect was due to membrane disturbance, as they found higher leakage of potassium and phosphate after LAE treatment of *E. coli*, *L. monocytogenes* and *Br. thermosphacta*. Zhao et al. [38] showed that LAE treatment of *E. coli* resulted in a higher release of nucleic acids and proteins, higher uptake of propidium iodide or N-phenyl-1-napthylamine as well as increased membrane depolarization. In a recent publication by Ma et al. [16], the authors stated that Gram-negative bacteria tend to be more resistant to LAE than the Gram-positive bacteria species, probably due to the specific external membrane of Gram-negative bacteria, as the Gram-negative membrane is primarily composed of lipopolysaccharides and phospholipids acting as a permeability barrier against external toxic compounds [16]. The missing effect of LAE on *S*. Typhimurium, inoculated on pork, on day 0 of storage might be due to this membrane difference. However, this missing effect should not be overestimated, as during further storage up to day 7, a significant difference between LAE treated and untreated pork, inoculated with *S*. Typhimurium, was shown.

The varying *Br. thermosphacta* results during MAP storage of the different meat species pig, cattle and chicken after the initial LAE dependent reduction on day 0 are difficult to discuss, as in beef the LAE effect persisted and in pork and chicken meat vanished. At first, it could be suggested that the varying LAE results during storage are principally related to matrix differences of the pork, beef and chicken meat and the accompanied varying growth properties of *Br. thermosphacta* during MAP storage. The latter assumption is supported by different studies that also stored (minced or chopped) meat in high oxygen MAP. For example, Cauchie et al. [39] showed that after inoculation of minced pork with *Br. thermosphacta* and storage at 4 °C in MAP (70% O_2_, 30% CO_2_) a slight increase could be obtained up to day 13 accompanied with a low maximal specific growth rate, whereas Koller et al. [40] showed, in a similar experiment with pork, a significant reduction of the *Br. thermosphacta* number up to day 14. Esmer et al. [41] found, in similar experiments with minced beef, at first an increase in the number of *Br. thermosphacta* followed by a decrease up to day 14, whereas Kamenik et al. [42] exhibited a slight increase in *Br. thermosphacta* on pork and beef after 14 days of MAP storage in 80% O_2_ and 20% CO_2_. Using the same gas mixture, Mastromatteo et al. [43] showed that during the MAP storage of patties, made in equal parts of chicken, turkey and ostrich meat, up to day 8, *Br. thermosphacta* initially increased and then decreased. Again, in the present study during MAP storage, the bacteria number of *Br. thermosphacta* decreases in pork, increases in chicken meat and increases in beef followed by a decrease during further storage. Anyway, the reasons for the varying effects of the meat matrix on the growth properties are difficult to determine, as not only the matrix, but also the conditions for the growth of *Br. thermosphacta* have to be considered. *Br. thermosphacta* is tolerant to high salt (up to 10% NaCl) and low pH conditions (down to 5.0) and is able to grow at refrigerator temperatures (down to 0 °C) [44,45]. The water activity (a_w_ value) should not be lower than 0.96 (Blickstad). Beside this, the bacteria species shows mainly no proteolytic activity and at 4 °C a reduced lipolytic activity, depending on the strain [46]. As meat has a high a_w_ value above 0.96, a low salt content and as the meat in the present study was stored at the same temperatures, it could be suggested that factors like the pH values or the carbohydrate content have to be considered with regard to the varying growth properties of *Br. thermosphacta* on the different meat species pig, cattle and chicken. With regard to the pH value, Goncalves et al. [47] showed that maximum growth rate of *Br. thermosphacta* at a pH of 5.5 was lower than at a pH of 6.0. This might explain that in chicken meat samples the bacteria number increases during MAP storage, as chicken meat has higher final pH values of around 5.96 [48]) compared to beef or pork with final pH values of around 5.5 [49] or 5.7 [50], respectively. With regard to the carbohydrate content, we considered publications that determined the (residual) concentrations of glycogen and glucose in the meat (at the final pH value, 24 h to 48 h p.m.). The content of these carbohydrates at this point of time is usually quite low, but also very variable. For example, the glycogen or glucose contents are 5 to 15 µmol/g or 3 to 9 µmol/g in chicken breast muscles [51,52,53], 10 to 20 µmol/g or 7 to 15 µmol/g in the porcine *M. longissimus lumborum* (LT) [54,55,56,57] and 4 to 16 µmol/g or 5 to 6 µmol/g in the bovine LT [58,59]. Immonen et al. [60] found residual glycogen values between 20 µmol/g and 60 µmol/g meat. However, despite the high variation of the results, an impact of the low carbohydrate content of the meat on the differing growth properties of *Br. thermosphacta* is not very likely. However, further investigations might be useful.

The missing effects of the LAE treatment on the color values directly after treatment and during MAP storage mainly agree with other publications analyzing (chicken) meat [9,18,22,25,61]. However, comparable studies with pork or beef have not been published, as far as our database research shows. As the color of meat is an important characteristic for the buying behavior of the consumers, the presented results can be considered positive. Meat color is influenced by several factors like genetic, age, sex, muscle type and is strongly related to the protein myoglobin (Mb) and the proportion of its redox forms OxyMb, DeoxyMb and MetMb [62]. In addition to proteolysis of Mb during meat ripening, important chemical steps to change the proportions of the Mb redox forms and thus the meat color are oxygenation of DeoxyMb to OxyMb and oxidation of OxyMb to MetMb. The first results in the change in the meat to a purplish-red or purplish-pink color and is often related to MAP storage of the meat at (high) oxygen concentrations. The oxidation to MetMb results in an alteration to a brownish color which might reduce the consumer acceptance [62]. As the (antimicrobial) impact of LAE is not related to oxidative alterations of the bacteria [33,38], the missing effect of the substance on the meat and its color is plausible also considering that, in the present study, the proportions of the Mb redox forms were also not influenced by the LAE treatment. The latter results agree with the study of Bechstein et al. [9] in chicken. Again, comparable studies in beef and pork with Mb redox form results have not been published.

The mainly missing effects of the starter culture bacteria species *L. sakei* and *S. carnosus* on the target bacteria species *Br. thermosphacta* on pork and chicken meat, also during MAP storage, agree on the hand with other studies that analyzed these effect using meat [12,13,63,64,65,66]. On the other hand, other studies found lower as well as higher *Br. thermosphacta* concentrations initially and during (MAP) storage after treatment of meat with starter culture bacteria [12,13,27,28,29,66], which agrees with the present results of beef on day 14. As the latter effect was found only on day 14 of storage, the results should be treated with caution. Interestingly, most of the studies on the impact of starter culture bacteria on the microbiological load of beef have been published. Although several studies support the present results, the varying results are difficult to discuss. This might be related, for example, to the meat species, starter culture bacteria species and strain or the target bacteria species (e.g., *Br. thermosphacta*) which differ in the publications. For example, Zhang et al. [66] found on vacuum-packaged beef a reduction of *Br. thermosphacta* using *L. sakei* and no effect on Brochothrix using *L curvatus* (SafePro B-LC-48), whereas Castellano et al. [29] found in a comparable study design clear reductions of *Br. thermosphacta* using a *L. curvatus* strain (CRL705). Xu et al. [11] showed in a literature overview that, in in vivo experiments, the starter culture bacteria species and their strain differently influence the results of *Br. thermosphacta* on beef and lamb meat. Jones et al. [65,67] showed in in vitro investigations that different starter culture bacteria species and their strains show varying antimicrobial activities on target bacteria species like *Br. thermosphacta*. Considering the influence of the target bacteria species/strain, Saraoui et al. [68] analyzed in in vitro experiments if *L. piscium* changes the growth of 42 *L. monocytogenes* strains. The authors showed that the inhibition/reduction varied from 0.7 up to 3.7 log_10_ cfu/mL. As this shows a clear impact of the *L. monocytogenes* strain, an impact of the strain could be also assumed for other bacteria species like *Br. thermosphacta*. Principally, starter cultures are used, for example, for the production of raw fermented sausages, due to their rapid acidification properties that influence the sensory quality [69]. This acidification is also important with regard to the microbiological safety of the product, as many bacteria species do not grow at low pH values. However, starter culture bacteria also influence microbiological contamination due to the competitive growth accompanied with competition to the meat nutrients and the production of bacteriocines and acids that specifically influence the growth of other bacteria species [69]. Therefore, an impact of the starter cultures, if applied as a preservative on the meat surface, is quite probable, but this was unfortunately not clearly proven with regard to *Br. thermosphacta* in the different studies [12]. Within one starter culture species the properties (e.g., bacteriocine production, acidification) can differ between the strains [69]. For example, Simsek et al. [70] presented that different starter culture species/strains vary with regard to their lactic acid production or their proteolytic or amylolytic activities which also influences their acidification properties and the competitive growth behavior.

As we found some significant reductions of the target bacteria after treatment with starter culture bacteria initially (*L. monocytogenes* on beef) or during storage (*S*. Typhimurium on pork), an antimicrobial effect of *L. sakei* and *St. carnosus* could be partly proven in contrast to the *Br. thermosphacta* results. Therefore, the principal discussion about the impact of the starter culture bacteria in the previous section (e.g., acidification, competitive growth, bacteriocines) should be also considered for these results. In many studies, *Pseudomonas* spp. and *Enterobacteriaceae* as target bacteria species were investigated [12,13,27,64,66,71], whereas comparable studies with *Salmonella* spp., *Campylobacter* spp. or *Listeria* spp. also have been published [13,27,28,65,72,73,74,75]. However, already published data are quite variable. For example, Ruby et al. [72] and Nikodinska et al. [75] treated ground beef and ground pork, respectively, with Lactobacilli showing no impact on *Salmonella* spp. during storage, whereas Olaoye et al. [73] and Yang et al. [13] showed a generally decreasing effect of lactic acid bacteria on S. Typhimurium, inoculated on goat meat and beef. Chailllou et al. [74] determined in q-RT-PCR experiments with ground beef a reduction of *S*. Typhimurium not initially after treatment but after four days of cold storage, like in the present study with pork. Like in the present study, Djenane et al. [27], Castellano et al. [28], Olaoye et al. [73] or Nikodinska et al. [75] also showed, after storage of beef, goat meat or ground pork with different lactobacilli strains, a clear reduction of *L. monocytogenes* or *L. innocua* during storage. The presented *C. jejuni* results on chicken meat disagree with the study of Jones et al. [65] who found on beef a significant reduction of this target species using different *L. sakei* strains. In this context it is again important to note that Jones et al. [65,67], Simsek et al. [70] or Wang et al. [76] show in in vitro investigations that different starter culture bacteria species and their strains show varying antimicrobial activities on target bacteria species like *L. monocytogenes* or *C. jejuni*. 

The missing effects of the starter culture treatment on the color values directly after starter culture treatment and during MAP storage mainly agrees with other publications that analyzed meat [27,64,66,71,72,77,78,79]. However, in some of these studies, a few effects of the treatment on the color results have been shown. For example, Ruby et al. [72] found at the end of storage on day 6 lower a* values of the beef. Lasziewiecz et al. [71] showed that *L. plantarum*-treated poultry batters had, on day 1, significantly higher L* values than the control samples but not during further storage up to day 7, whereas *L. brevis*-treated batters had only on day 4 of storage significantly lower L* values compared to the control samples. However, the color results of the latter study have to be considered with care, as nitrite-containing batters were used and an effect of the starter culture bacteria on the curing process could not be excluded. In contrast to the present study, Xu et al. [12] and Yang et al. [13] found significant effects of the treatment of pork batters, lamb meat and beef, respectively, with starter culture bacteria on the L* a* b* results of the treated samples. In the study by Xu et al. [12], the L* and b* results were significantly higher and the a* values lower, if the meat was treated either with *L. sakei*, or with *L. sakei* and *St. carnosus*, independent of storage up to day 20. Yang et al. [13] found higher L* values, independent of the storage time, and higher a* and b* results on day 12 and days 9 and 12 of storage, respectively, if the meat was treated with *L. sakei*. However, the present results indicate that the addition of starter culture bacteria like *L. sakei* and *St. carnosus* has no impact on the meat color which again is very important with regard to the consumer acceptance. As already stated above, the meat color is mainly influenced by the Mb concentration and by the percentages of the Mb redox forms [62]. Therefore, the myoglobin redox form results in the present study clearly agree with the color values. Both parameters were not influenced by the treatment with *L sakei* and *St. carnosus*. However, comparable studies that analyzed these parameters are quite rare. The present results are supported by Djenane et al. [27] who also did not determine an impact of the starter culture bacteria treatment on the MetMb results during storage of beef. Interestingly, the authors also did not find an impact on the color values of beef.

## 5. Conclusions

The microbiological results of the combination of LAE and starter culture bacteria in the present study show that in nearly all experiments initially and during MAP storage no additive effect of both preservations methods could be obtained and that the combinatory effects were mainly due to the impact of LAE and less to the impact of the starter culture bacteria. Exceptions were the significant reducing effects of the combination on the bacteria number of *Br. thermosphacta* on pork and beef on day 7 (pork) and day 14 (beef), respectively, and of *S*. Typhimurium on pork on days 0 and 7, which indicate small additive effects. However, the data indicate that it is possible to reduce the microbiological load of meat by treating it with LAE and starter culture bacteria in combination, thereby improving the food safety. However, before combined application, the effectiveness of this procedure should be further evaluated, to prevent unnecessary treatments and costs. In this context it might be necessary to adjust the starter culture bacteria species and/or strains. Beside this, it is important that the single and combined application in the present study did not affect the color and myoglobin redox form percentage results of the treated meat and it is therefore not an exclusion criterion for the use of the treatment of the meat.

## Figures and Tables

**Figure 1 foods-12-04138-f001:**
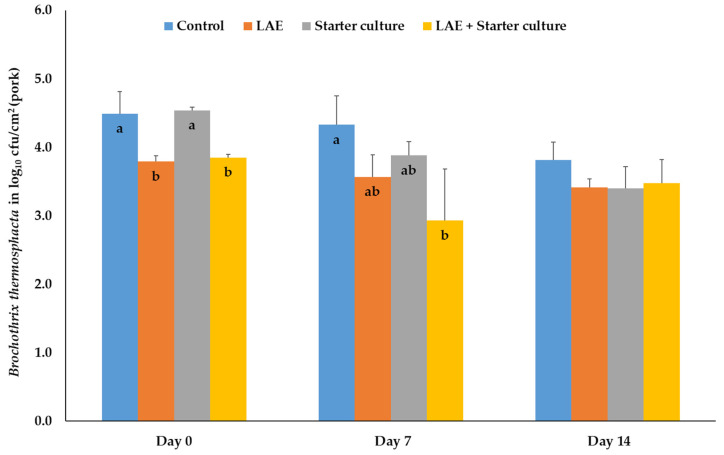
Mean and standard deviation values of the bacteria number of *Brochothrix thermosphacta*, inoculated on pork, depending on the day of storage in modified atmosphere (70% O_2_; 30% CO_2_) and the treatment after inoculation; Control = NaCl (0.9%) + Glycerol (3600 ppm); LAE = NaCl (0.9%) + LAE (4000 ppm); Starter culture = *Staphylococcus carnosus* and *Lactobacillus sakei* (10^6^ cfu) + Glycerol (3600 ppm); LAE + Starter culture = LAE (4000 ppm) + *Staphylococcus carnosus* and *Lactobacillus sakei* (10^6^ cfu); ^a,b^ columns with different letters on the same storage day differ significantly (*p* ≤ 0.05).

**Figure 2 foods-12-04138-f002:**
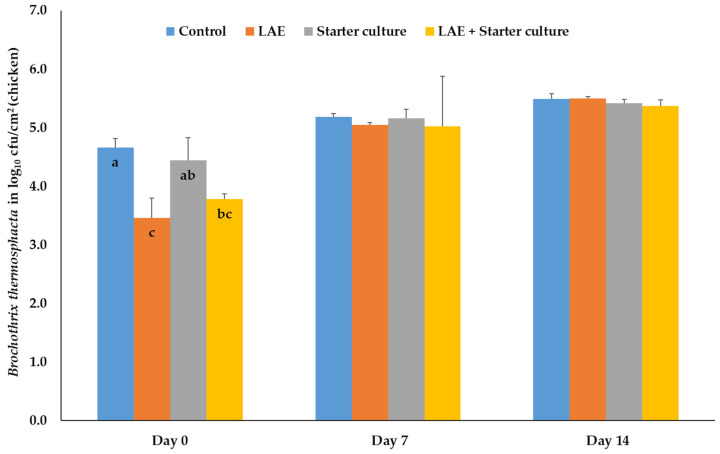
Mean and standard deviation values of the bacteria number of *Brochothrix thermosphacta*, inoculated on chicken meat, depending on the day of storage in modified atmosphere (70% O_2_; 30% CO_2_) and the treatment after inoculation; Control = NaCl (0.9%) + Glycerol (3600 ppm), LAE = NaCl (0.9%) + LAE (4000 ppm); Starter culture = *Staphylococcus carnosus* and *Lactobacillus sakei* (10^6^ cfu) + Glycerol (3600 ppm); LAE + Starter culture = LAE (4000 ppm) + *Staphylococcus carnosus* and *Lactobacillus sakei* (10^6^ cfu); ^a,b,c^ columns with different letters on the same storage day differ significantly (*p* ≤ 0.05).

**Figure 3 foods-12-04138-f003:**
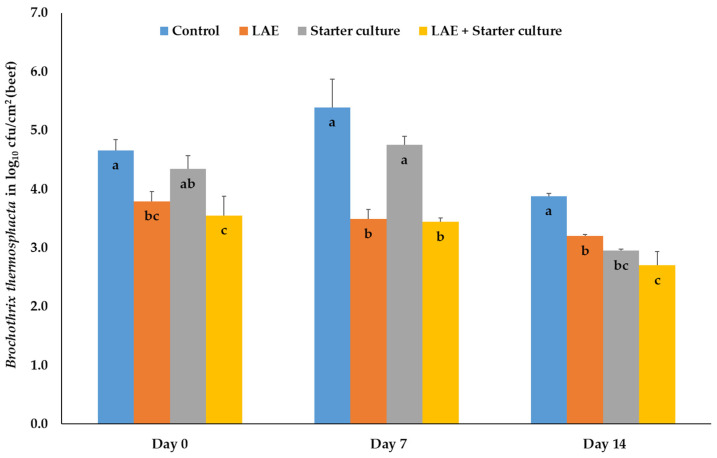
Mean and standard deviation values of the bacteria number of *Brochothrix thermosphacta*, inoculated on beef, depending on the day of storage in modified atmosphere (70% O_2_; 30% CO_2_) and the treatment after inoculation; Control = NaCl (0.9%) + Glycerol (3600 ppm), LAE = NaCl (0.9%) + LAE (4000 ppm); Starter culture = *Staphylococcus carnosus* and *Lactobacillus sakei* (10^6^ cfu) + Glycerol (3600 ppm); LAE + Starter culture = LAE (4000 ppm) + *Staphylococcus carnosus* and *Lactobacillus sakei* (10^6^ cfu); ^a,b,c^ columns with different letters on the same storage day differ significantly (*p* ≤ 0.05).

**Figure 4 foods-12-04138-f004:**
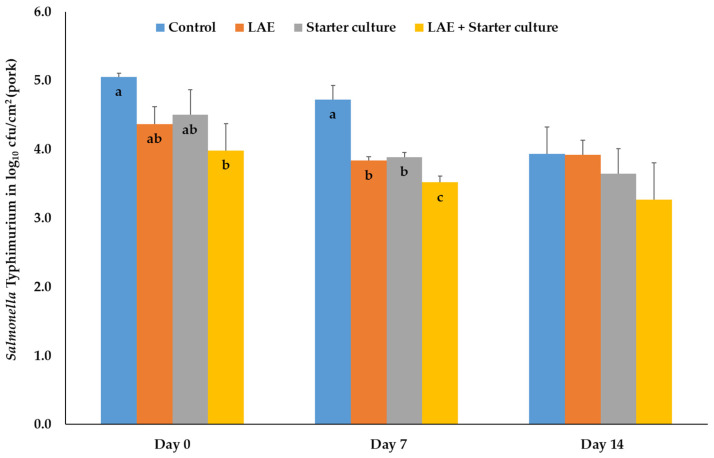
Mean and standard deviation values of the bacteria number of *Salmonella* Typhimurium, inoculated on pork, depending on the day of storage in modified atmosphere (70% O_2_; 30% CO_2_) and the treatment after inoculation; Control = NaCl (0.9%) + Glycerol (3600 ppm), LAE = NaCl (0.9%) + LAE (4000 ppm); Starter culture = *Staphylococcus carnosus* and *Lactobacillus sakei* (10^6^ cfu) + Glycerol (3600 ppm); LAE + Starter culture = LAE (4000 ppm) + *Staphylococcus carnosus* and *Lactobacillus sakei* (10^6^ cfu); ^a,b,c^ columns with different letters on the same storage day differ significantly (*p* ≤ 0.05).

**Figure 5 foods-12-04138-f005:**
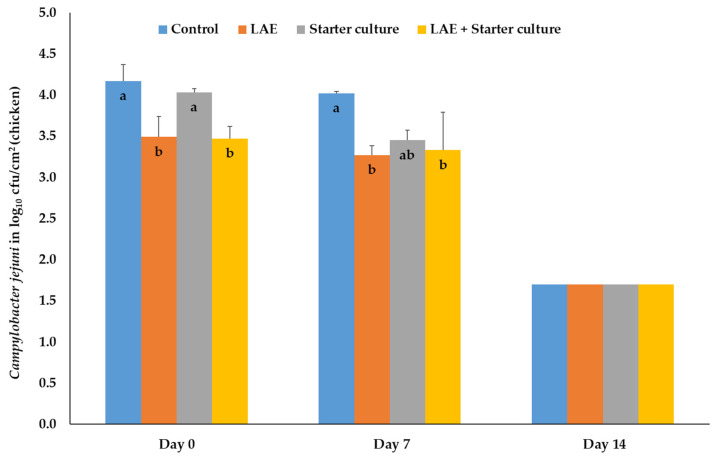
Mean and standard deviation values of the bacteria number of *Campylobacter jejuni*, inoculated on chicken meat, depending on the day of storage in modified atmosphere (70% O_2_; 30% CO_2_) and the treatment after inoculation; Control = NaCl (0.9%) + Glycerol (3600 ppm); LAE = NaCl (0.9%) + LAE (4000 ppm); Starter culture = *Staphylococcus carnosus* and *Lactobacillus sakei* (10^6^ cfu) + Glycerol (3600 ppm); LAE + Starter culture = LAE (4000 ppm) + *Staphylococcus carnosus* and *Lactobacillus sakei* (10^6^ cfu); ^a,b^ columns with different letters on the same storage day differ significantly (*p* ≤ 0.05).

**Figure 6 foods-12-04138-f006:**
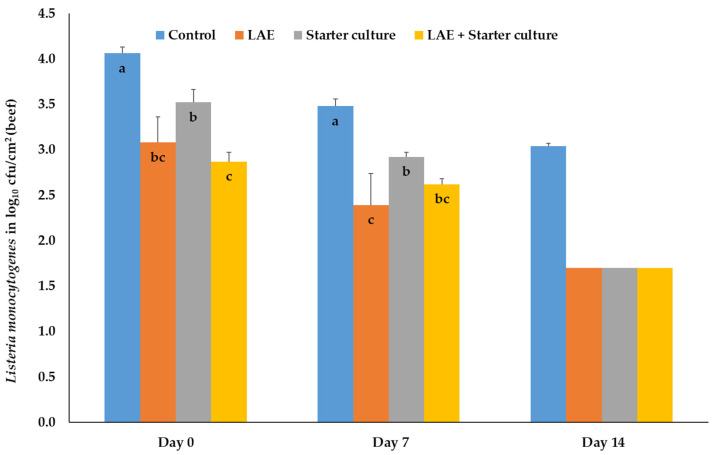
Mean and standard deviation values of the bacteria number of *Listeria monocytogenes*, inoculated on beef, depending on the day of storage in modified atmosphere (70% O_2_; 30% CO_2_) and the treatment after inoculation; Control = NaCl (0.9%) + Glycerol (3600 ppm), LAE = NaCl (0.9%) + LAE (4000 ppm); Starter culture = *Staphylococcus carnosus* and *Lactobacillus sakei* (10^6^ cfu) + Glycerol (3600 ppm); LAE + Starter culture = LAE (4000 ppm) + *Staphylococcus carnosus* and *Lactobacillus sakei* (10^6^ cfu); ^a,b,c^ columns with different letters on the same storage day differ significantly (*p* ≤ 0.05).

**Table 1 foods-12-04138-t001:** Mean and standard deviation values of the color results depending on the treatment and the day of storage in modified atmosphere (70% O_2_; 30% CO_2_).

Treatment	Day of Storage	Lightness L*		Redness a*		Yellowness b*	
		Mean	SD	Mean	SD	Mean	SD
Pork							
Control	0	57.0	1.4	8.2	2.0	8.0	1.6
LAE	0	56.4	1.9	9.0	2.9	8.6	1.4
Starter	0	54.4	4.3	8.1	1.8	7.3	0.6
LAE + Starter	0	55.3	4.3	8.4	2.1	7.8	0.6
Control	14	58.8	2.3	7.9	2.0	9.8	1.6
LAE	14	59.2	1.0	7.7	2.1	10.2	2.4
Starter	14	59.8	1.7	8.2	1.5	10.2	1.3
LAE + Starter	14	60.2	2.0	7.1	1.7	10.0	1.2
Chicken meat							
Control	0	57.0	5.4	2.4	0.3	6.9	2.3
LAE	0	58.0	2.3	2.0	1.0	7.9	0.5
Starter	0	57.3	3.9	2.6	0.4	7.4	2.1
LAE + Starter	0	57.5	4.5	2.2	0.2	7.6	1.5
Control	14	64.4	1.2	0.9	0.4	8.8	0.7
LAE	14	63.8	2.5	0.9	0.3	8.1	2.1
Starter	14	65.1	1.6	0.7	0.4	9.1	1.1
LAE + Starter	14	64.4	2.6	0.7	0.1	8.9	1.4
Beef							
Control	0	38.8	5.0	22.0	0.6	11.8	1.1
LAE	0	40.1	3.2	22.3	2.1	12.1	0.3
Starter	0	42.5	4.5	23.2	0.6	12.9	0.8
LAE + Starter	0	40.3	2.1	22.3	0.8	12.4	0.5
Control	14	40.5	3.1	8.9	1.1	11.4	0.5
LAE	14	39.1	1.7	10.5	3.2	10.7	0.3
Starter	14	43.7	4.9	9.9	5.3	11.5	0.4
LAE + Starter	14	40.7	3.7	11.0	7.6	11.4	0.5

Control = NaCl (0.9%) + Glycerol (3600 ppm); LAE = NaCl (0.9%) + LAE (4000 ppm); Starter = *Staphylococcus carnosus* and *Lactobacillus sakei* (10^6^ cfu) + Glycerol (3600 ppm); LAE + Starter = LAE (4000 ppm) + *Staphylococcus carnosus* and *Lactobacillus sakei* (10^6^ cfu).

**Table 2 foods-12-04138-t002:** Mean and standard deviation values of the percentages of Oxy-Myoglobin (OxyMb) and Met-Myoglobin (MetMb) depending on the treatment and the day of storage in modified atmosphere (70% O_2_; 30% CO_2_).

Treatment	Day of Storage	OxyMb (%)		MetMb (%)	
		Mean	SD	Mean	SD
Pork					
Control	0	33.1	3.0	49.5	2.0
LAE	0	31.9	2.2	50.5	1.5
Starter	0	35.5	2.8	47.4	2.3
LAE + Starter	0	32.9	1.8	49.8	1.7
Control	14	31.2	5.7	49.9	5.5
LAE	14	30.6	2.4	50.3	2.9
Starter	14	29.7	2.4	50.9	2.0
LAE + Starter	14	28.2	2.2	52.5	2.8
Chicken meat					
Control	0	19.5	1.8	59.4	1.8
LAE	0	17.9	3.6	60.9	2.8
Starter	0	17.4	1.8	61.2	1.4
LAE + Starter	0	19.7	2.4	59.5	1.9
Control	14	12.4	2.0	65.2	1.5
LAE	14	13.0	1.7	64.6	2.0
Starter	14	13.3	1.1	64.3	1.2
LAE + Starter	14	13.7	1.8	63.9	2.3
Beef					
Control	0	68.5	3.4	25.3	3.2
LAE	0	66.3	6.2	27.5	6.0
Starter	0	62.9	1.3	31.7	1.6
LAE + Starter	0	61.7	5.7	33.0	4.9
Control	14	13.4	4.3	81.1	5.7
LAE	14	15.8	9.0	79.2	9.0
Starter	14	17.1	17.6	77.9	17.0
LAE + Starter	14	18.4	21.1	76.4	20.4

Control = NaCl (0.9%) + Glycerol (3600 ppm); LAE = NaCl (0.9%) + LAE (4000 ppm); Starter = *Staphylococcus carnosus* and *Lactobacillus sakei* (10^6^ cfu) + Glycerol (3600 ppm); LAE + Starter = LAE (4000 ppm) + *Staphylococcus carnosus* and *Lactobacillus sakei* (10^6^ cfu).

## Data Availability

Data is contained within the article.

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
