# Peer review of "Investigation of the Suitability of a Combination of Ethyl-Να-dodecanyl-L-arginat_HCl (LAE) and Starter Culture Bacteria for the Reduction of Bacteria from Fresh Meat of Different Animal Species"

_foods, 2023, doi:10.3390/foods12224138_

Round 1

Reviewer 1 Report

Comments and Suggestions for Authors

This manuscript reported the effect of LAE and starter culture bacteria during the storage of different animal meat. As stated by the authors themselves, combination effects were limited. And the results obtained in this study were not conclusive and were just among the reports on this controversial subject. However, this manuscript includes many informative findings.

Specific comments:
1. Title: LAE is not a common abbreviation. Uncommon abbreviations are not good for use in the title.

2. Page 3 Line 102: approximately 106 cfu of a mixture (1:1) of Staphylococcus carnosus (DSM 20501) and Lactobacillus sakei (DSM 6333)

Does it mean 106 cfu in total or 106 cfu of each species of bacteria?

3. Page 3 Line 114: Periods should not be used for separating digits. 100,000 ppm

4. Page 4 Line 173: Bacterial number of swapping on the meat surface should be expressed by cfu / cm2. (The authors used cfu / cm2 in the latter part of the text.)

5. Page 4 Line 173: I cannot understand why half of the detection limit was log10 1.7 cfu/ g.

6. Page 5 Figure 1: The number of Brochothrix thermosphacta seemed to increase slightly in LAE + Starter culture group at Day 14 compared with Day 7. Don't you think it is necessary to compare among the same sample at different storage days, (or at least to compare with Day 0), statistically? The same applies to other figures.

Comments on the Quality of English Language

N/A

Author Response

Please consider the attached file.

Reviewer 2 Report

Comments and Suggestions for Authors

Dear authors,

The manuscript entitled „Investigation of the suitability of a combination of LAE and starter culture bacteria for the reduction of bacteria from fresh meat of different animal species“ reflect interesting an useful research with great potential for the future.

Considering that annual losses amount to approximately 20% of the initial meat production, these losses primarily stem from spoilage, which leads to a decline in the sensory quality of meat products. Various mechanisms contribute to these spoilage defects, including natural processes like lipid oxidation and autolytic enzymatic reactions in muscle cells post-slaughter. Nevertheless, the primary culprit behind spoilage is the unavoidable contamination by microorganisms, particularly bacteria, during the processing of animals into meat products, and their subsequent growth and metabolic activities during storage. Each processing step can influence microbial contamination, and storage conditions can shape the structure of bacterial communities, thereby influencing the occurrence of microbial spoilage over time. Hence, it is imperative that research efforts focus on strategies to prevent and reduce meat spoilages.

Therefore, the authors of this paper investigatated application of LAE (ethyl-Nα-dodecanyl-L-arginate hydrochloride) and the starter culture bacteria Staphylococcus carnosus and Lactobacillus sakei, single, or in combination influences the bacteria number on pork, chicken meat and beef, inoculated with Brochothrix (Br.) thermosphacta (all meat species) or Salmonella (S.) Typhimurium (pork), Campylobacter (C.) jejuni (chicken) and Listeria (L.) monocytogenes (beef), before packaging under modified atmosphere and on days 7 and 14 of storage.

The authors showed that there was no additive effect between the combination of LAE and starter culture bacteria. Antimicrobial efficacy was prescribed mainly due to the impact of LAE and less due to the impact of the starter culture bacteria. The obtained data suggests that it is possible to reduce the microbiological load of meat by treating it with LAE and starter culture bacteria in combination, thereby improving food safety.

The authors emphasized the need for further evaluation of this procedure before its combined application to prevent unnecessary treatments and costs. This research indicates a promising avenue for improving meat safety, potentially benefiting both consumers and the food industry. The combination of LAE and starter culture bacteria may offer an innovative approach to enhancing the quality and safety of meat products. Further studies and trials are required to validate the effectiveness and safety of this approach before it can be widely implemented in meat processing. The work is well-written, concise, and, in my opinion, has enough information in all sections: abstract, introduction, material and methodology, discussion, and conclusion.

Overall, the research in the manuscript is fairly well organized and carried out. Both the science and the presentation are strong. So, I would suggest minor corrections. The manuscript requires minor English language corrections.

Line 98: The meat samples were inoculated with approximately 105 colony-forming units (cfu); please define how the inoculum was prepared. Please add equivalence to the McFarland standards unit.

Line 108: “colony forming units“ should be deleted; the authors introduced the meaning of abbreviation in line 98.

Line 110: cfu/ml should be written without space. Check and correct all manuscript.

Line 114: and Line 115: The percentage after the number should be without space (please correct through all manuscript).

Line 125: (MAP, 70 % O2, 30 % CO2) % after number should be without space (please correct through all manuscript).

Line 152: at 4 °C, °C after number should be without space (please correct through all manuscript).

Comments on the Quality of English Language

 The manuscript requires minor English language corrections.

Author Response

Please consider the attached file.
